# *BCOR*, *BCORL1*, and *BCL6* Mutations in Pediatric Leukemias

**DOI:** 10.3390/cancers17152443

**Published:** 2025-07-23

**Authors:** Thomas C. Fisher-Heath, Aastha Sharma, Mark S. Marshall, Tiffany Brown, Sandeep Batra

**Affiliations:** 1Department of Pediatrics, Indiana University School of Medicine, Riley Children’s Health Indiana University Health, Indianapolis, IN 46202, USAmmarsha@iu.edu (M.S.M.); tbrown34@iuhealth.org (T.B.); 2Indiana University Melvin and Bren Simon Cancer Center, Indiana University School of Medicine, Indianapolis, IN 46202, USA

**Keywords:** acute myeloid leukemia, myelodysplastic syndrome, *BCL6*, *BCOR*, *BCORL1*

## Abstract

Genetic changes play an important role in the development of acute myeloid leukemia (AML) and myelodysplastic syndrome (MDS). Some rare genetic mutations, specifically in the *BCOR* and *BCORL1* genes, have been found in adult patients, but have not been well described in pediatric patients. In this report, we share a case series of pediatric and adolescent patients who have these specific mutations. While this helps shed light on a rare finding, more research with larger groups of patients is needed to truly understand what these mutations mean for young people with AML or MDS.

## 1. Introduction

Several rare genetic alterations in pediatric leukemias impacting risk stratification and treatment have recently been identified using next-generation sequencing (NGS). NGS is a pivotal tool that allows comprehensive profiling, the identification of genetic mutations and epigenetic changes, and targeted treatment [1].

B-cell lymphoma-6 corepressor (BCOR) is a transcription factor, located on chromosome X, and plays a key role in hematopoiesis and stem cell function and pluripotency, and mutations can lead to hematopoietic malignancies such as de novo and secondary AML [2,3].

*BCOR* and its homolog, B-cell lymphoma-6 corepressor-like protein 1 (*BCORL1*), are rarely altered (<2% incidence) in pediatric leukemias [4,5,6,7]. Similarly, BCOR mutations rarely occur in adult leukemias, with one study reporting BCOR mutations in 4% of 262 adults with cytogenetically normal acute myeloid leukemia (CN-AML) [4]. In adults, loss-of-function mutations in *BCOR* are associated with poor prognoses, but the prognostic significance of *BCORL1* has yet to be examined due to its relative rarity [5].

*BCOR* and *BCORL1* are corepressors of the B-cell lymphoma-6 (*BCL6*) gene (Figure 1) [6] that function as a part of the non-canonical PRC1.1 complex, responsible for the repression of several antitumor genes through chromatin modifications and opposition of the differentiation of cells toward myeloid lineage, through the repression of *HOX* and *Cepb* family genes [4,8,9]. The disruption of this complex, e.g., *BCOR* loss of function, leads to multiple downstream effects, which include the promotion of clonal expansion, persistence of DNA damage, and absence of cellular differentiation, contributing to leukemogenesis [10,11,12].

The role of BCOR in promoting leukemogenesis is further elucidated by several murine models. *P53* knockout mice harboring the *BCOR* exon 4 deletion (*BCOR*^ΔE4/y^) develop T-cell acute lymphoblastic leukemia (T-ALL) in a NOTCH1-dependent manner [13]. *BCOR*^−/−^*DNMT3A*^−/−^ double-knockout mice develop an acute erythroid leukemia (AEL) phenotype [14]. Frameshift *BCOR* mutations in a nine-base-pair hotspot in exon 8 collaborate with oncogene *NUP98-PHF23* (*NP23*) to generate an aggressive acute lymphoblastic leukemia of B-1 lymphocyte progenitor origin (pro-B1 ALL) [15].

Mice harboring only *BCOR* mutations do not develop leukemia, but have increased circulating peripheral blood neutrophils, without significant changes in lymphocyte, platelet, or erythrocyte counts [6]. *BCOR* knockout mice crossed with oncogenic *KRAS*-variant mice develop a lethal leukemic phenotype [14]. Notably, BCOR-/KRAS-variant mice have significantly worse survival than those with the oncogenic *KRAS* mutation alone [14]. In summary, these murine models support the notion that *BCOR/BCORL1* mutations contribute to leukemogenesis but are likely insufficient to induce leukemogenesis alone.

Spliceosome mutations are found in the majority of MDS patients and are frequently the earliest mutations that are identified. Additional subclonal somatic lesions are acquired subsequently and often confer a resistant phenotype and drive the progression from MDS to AML [16]. Bernard et al. performed genomic profiling of 3233 patients with adult MDS and identified 3.5% (*n* = 114) of patients as having mutations in BCOR (85%), 33% as having those in BCORL1, and 17% as having those in both genes. Additionally, subclonal RUNX1 mutations were common (41%) in these patient samples. BCOR/L1-mutated MDS was characterized by severe thrombocytopenia, high blast counts, a shorter OS (median, 2.2 years), and a 24% 2-year incidence of AML transformation [17].

BCOR mutations confer an adverse prognosis in adult AML and were incorporated into the 2022 European Leukemia Net (ELN) classification as an adverse risk marker [18,19]. The impact of BCOR/BCORL1 mutation on the outcomes of AML/MDS patients that have received an allogeneic hematopoietic stem cell transplant has not been studied extensively [20].

The genetic landscape of pediatric AML differs from that of adult AML. Adult patients with AML often display increased mutational burden and fewer cytogenetic alterations when compared to children [21]. Currently, genetic abnormalities in pediatric AML are classified into three categories based on their prognostic significance: favorable risk, intermediate risk, and adverse risk. Favorable-risk genetic alterations include core-binding factor protein fusions like *RUNX1:RUNX1T1*, *CBFB:MYH11*, bi-allelic *CEBPA* mutations, and *NPM1* mutations without *FLT3-ITD*. Favorable-risk genetic alterations portend better prognosis with 5-year overall survival rates surpassing 80%.

In contrast, adverse genetic changes have significantly worse prognosis, with 5-year overall survival rates less than 40% in some cases [22]. *FLT3-ITD* mutations occur in 20–25% of pediatric AMLs and are among the most clinically significant adverse alterations [23]. The prognosis associated with *FLT3-ITD* mutations varies based on the allelic ratio, with higher allelic ratios leading to worse prognosis [23]. Other well-established adverse genetic alterations include *NUP98* gene fusions, *RAS* pathway alterations (*NRAS*, *KRAS*, *PTPN11*), and *CSF3R* mutations [24,25]. Additionally, karyotypic changes such as monosomy 7 and monosomy 5/del5q are associated with worse prognosis and lead to a rare subtype of AML, acute megakaryoblastic leukemia (AMKL). *KMT2A* rearrangements, formerly *MLL* rearrangements, represent an emerging adverse genetic change [26]. This rearrangement has over one hundred described fusion partners, but is more commonly observed in infants less than one year of age [21,27].

In between favorable- and adverse-risk genetic changes fall intermediate-risk mutations. There is no set number of intermediate-risk mutations [28]. By definition, these mutations encompass all those genetic changes that have not been well described as either favorable or adverse [29]. The prognosis for this group varies significantly based on concurrent mutations, and overall survival is near 50% [28].

Here, we review a case series of pediatric and adolescent patients, with de novo AML/MDS, harboring *BCOR/BCORL1* mutations, adding to the growing body of research investigating the role of *BCOR* and *BCORL1* mutations in pediatric myeloid malignancies.

## 2. Materials and Methods

We identified a cohort of patients diagnosed with acute lymphoblastic leukemia (ALL), acute myeloid leukemia (AML), or myelodysplastic syndrome (MDS) at Riley Hospital for Children between January 2015 and December 2023 who had next-generation sequencing (NGS) performed on their bone marrow aspirate samples, utilizing the FoundationOne Heme (https://www.foundationmedicine.com/test/foundationone-heme, accessed on 1 January 2025) assay, and harbored mutations in *BCL6*, *BCOR*, and *BCOR1*, at diagnosis and/or relapse, and conducted a retrospective chart review. FoundationOne Heme utilizes DNA sequencing to interrogate the entire coding sequence of 406 oncogenes, selected introns of 31 genes involved in rearrangements, and the RNA sequencing of 265 genes known to be somatically altered in hematological malignancies (https://www.foundationmedicine.com/test/foundationone-heme, accessed on 1 January 2025). Relapse-free survival (RFS) was defined as survival without evidence of disease recurrence at three years off therapy. Kaplan–Meier survival curves were generated using MedCalc Software Limited© v22, MedCalc Software Ltd., Ostend, Belgium.

*BCOR*-like mutation variants were categorized by the type of mutation: intron variant, missense, gain of function, and synonymous. We queried large genetic databases [Catalogue of Somatic Mutations in Cancer (COSMIC) (https://cancer.sanger.ac.uk/cosmic, accessed on 1 January 2025), Clinical Knowledgebase Boost (CKB) (https://ckbhome.jax.org), the LifeOmic Precision Health Cloud (PHC) (https://www.medigy.com/offering/lifeomic-precision-health-cloud/, accessed on 16 July 2025), Online Mendelian Inheritance in Man (OMIM) (https://www.omim.org/), and Uniprot (https://www.uniprot.org/)] for each variant identified in our cohort of patients, to evaluate the likelihood of pathogenicity of each mutation. Variants were identified as potentially pathogenic if they were predicted to cause protein truncation. Additionally, using FoundationHeme, we identified co-mutations that occurred with *BCOR* and *BCORL1* mutations.

## 3. Results

A total of 102 patients with acute lymphoblastic leukemia (ALL) and 82 patients with AML/MDS had FoundationOne Heme NGS performed during their treatment course on their marrow samples. Eight (4.3%) patients harbored *BCOR, BCORL1*, or *BCL6* at diagnosis and/or relapse. In the ALL patient cohort, all mutations occurred at diagnosis (*n* = 3, 2.9%). In AML/MDS patients, mutations were identified at diagnosis (*n* = 2, 2.4%) and only at relapse (*n* = 3, 3.7%, but not at diagnosis) (Figure 2). Of the seventeen variants (*BCOR n* = 5, *BCORL1* n = 5, *BCL6 n* = 7), we identified three variants (18%) as possibly pathogenic (R1532fs, V1687fs, E1655fs) in the AML/MDS cohort (Figure 2). Additionally, we identified 95 co-mutations in these patients (Figure 3). *WT1* (*n* = 5)*, RUNX1* (*n* = 5), and *KRAS* (*n* = 4) mutations were the most common co-mutations identified among AML/MDS patients (Figure 3).

The three patients with ALL harboring *BCOR* mutations, identified at diagnosis, had excellent RFS (100% were relapse-free at three years). The RFS for patients with AML/MDS (*n* = 5) harboring *BCOR*/*BCORL1* mutations was only 20% (Figure 3), with the patient with MDS being the lone survivor. All AML/MDS patients received a stem cell transplant either as a part of their upfront consolidative therapy (*n* = 1) or for relapsed disease (*n* = 4).

## 4. Discussion

*BCOR*/*BCORL1* mutations occur infrequently in pediatric AML/MDS [4]. Only 6.1% of our AML/MDS patient bone marrow samples harbored these mutations. It is important to note that in three patients of the AML/MDS cohort, BCOR-like mutations were only identified at relapse. This finding indicates that BCOR-like mutations may be acquired in pediatric AML during disease evolution, persistence, and progression. The four AML patients died of progression despite receiving a stem cell transplant (Figure 3), with only one patient (MDS, patient #5) alive at the last-known follow-up.

Our findings concur with the published literature in adults, which has reported *BCOR* mutations to frequently occur in exon 4 (Figure 4) [30,31,32]. However, the mutations in exon 4 in our patient cohort are unlikely to be pathogenic, as they are synonymous mutations, and do not occur in known mutational hotspots. Three of the seventeen (17.6%) frameshift mutations identified in AML patients #1 and #6 (Figure 2) are potentially pathogenic, as they likely lead to truncations in the C-terminus of *BCOR* (R1532fs) and *BCORL1* (V1687fs, E1655fs) proteins, affecting the binding of *BCOR* with the *PRC*1.1 complex through its PUFD domain at the C-terminal end [6,31]. A truncation at the C-terminal end could prevent adequate binding and ultimately impair the function of the non-canonical *PRC*1.1 complex. Additionally, we identified two mutations (V1687fs, E1655fs) in the *BCORL1* gene affecting the *LXXLL* nuclear receptor recruitment motif on the C-terminal end (Figure 2 and Figure 4). Previous studies have identified that *BCORL1* mutations resulting in the absence of the *LXXLL* nuclear receptor impaired *BCORL1* function [33]. We predict that frameshift mutations in this region resulting in protein truncation would similarly impair *BCORL1* function.

In summary, our findings are similar to those of previously published studies that have reported that the distribution of BCOR/BCORl1 mutation sites are varied, often within the same patient sample, and that frameshift, missense mutations are common. The prognostic impact of the location or type of mutation continues to be an area of research that is evolving [15,34]. It appears that frameshift mutations may be associated with an inferior outcome in patients with adult hematopoietic malignancies compared to other type of mutations [34,35]. Due to the limited number of patients in our study, and the rarity of BCOR mutations in pediatric myeloid leukemias, we cannot draw definitive conclusions.

When examining co-occurring mutations, we found that most identified co-mutations were variants of unknown significance (VUSs) (Figure 3). *WT1* (patients #2, 4, and 6; no ALL patient)*, RUNX1* (patients #1, 6), and *KRAS* (patient #1) co-mutations were identified in our AML/MDS cohort of patients co-harboring *BCOR/BCORL1/BCL6* mutations (Figure 3). WT1 is often overexpressed in AML, CML blast crisis, and ALL [36]. Additionally, several studies have identified somatic mutations in WT1 in AML [37]. Wt-1 alterations are associated with chemo-resistance, increased relapse risk, and poor survival [38,39]. Similarly, alterations in RAS in AML/MDS are heterogeneous, differentially impact outcomes, and are likely influenced by the presence of concurrent mutations such as BCOR, leading to resistance to chemotherapy, venetoclax, azacitidine, and targeted therapy such as tyrosine kinase and FLT3 inhibitors [40,41].

Germline RUNX1 mutations cause Familial platelet disorder with associated myeloid malignancies (FPDMM), which is characterized by an increased risk of developing hematologic malignancies. Interestingly, in terms of frameshift mutations, *BCOR* is the most frequently altered gene in this patient cohort [42].

The presence of these co-mutations underscores the complexity of dynamic changes that can occur during the clonal expansion and transformation of pediatric AML/MDS. While the clinical significance of these findings is unclear, larger studies would perhaps help clarify their role.

Mutations in *BCL6*, *BCOR*, and *BCORL1* can contribute to leukemogenesis through various mechanisms and can be targeted with novel therapeutic strategies. *BCL6* has been the most well-researched target in this setting, as it is crucial for the survival and self-renewal of AML cells [43]. High *BCL6* expression in ALL and AML has been associated with treatment resistance and disease progression [10]. Zhang et al. recently identified WK499, a small-molecule compound that disrupts the interaction of *BCL6* with corepressor proteins [44]. This disruption causes downstream changes in the effects of *BCL6* and ultimately induces cell cycle arrest and apoptosis in AML cells [44]. Novel BCL6-targeting proteolysis-targeting chimeras (PROTACs) with effective antitumor activities against DLBCL in vitro and in vivo have been identified, and could be tested in human trials in the future [45]. In addition to its own anti-proliferative effects, *BCL6* inhibition has been shown to augment the anti-proliferative effects of classical chemotherapeutic agents like cytarabine in in vivo studies, making it a promising adjunct therapy [10].

There is also a precedent for disrupting genes that target *BCL6*, as BCOR and BCORL1 do. In diffuse large B-cell lymphoma, small-molecule inhibition of *STAT3*, a transcriptional target of *BCL6*, has shown promise in chemotherapy-refractory activated B-cell-like DLBCL [46].

Menin interacts with *KMT2A* directly and regulates *KMT2A* target genes. Several clinical trials have established small-molecule menin inhibitors as novel agents that impact resistant myeloid leukemias, with promising results [38]. Menin inhibitors display efficacy in early therapy of AML, but resistance can develop through epigenetic reactivation of non-canonical menin targets [47]. Zhou et al. showed that the loss of PRC1.1 subunits, like *BCOR* and *EZH*, led to menin resistance, but with functional PRC1.1 complexes, AML cells remained sensitive to menin inhibitors [47]. Interestingly, they found that the loss of PRC1.1 sensitizes AML cells to BCL2 blockade. These findings underscore the hypothesis that *BCOR/BCORL1* mutation-induced menin resistance may be overcome with bcl-2 inhibitors such as venetoclax [38]. While there are no current agents that target the PRC1.1 complex, inhibitors of *EZH1/2* and *EED*, subunits of the PRC2 complex, could also be novel effective downstream therapeutic targets [9].

## 5. Conclusions

This report demonstrates that BCOR-like somatic mutations are rare in pediatric leukemias and may or may not be pathogenic. Larger cooperative group studies are needed to further investigate the prognostic impact of BCOR/BCORL1 mutations in pediatric AML and MDS.

## Figures and Tables

**Figure 1 cancers-17-02443-f001:**
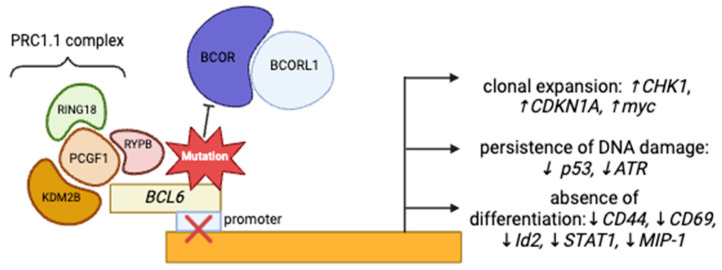
Figure describing the mechanisms through which BCOR/BCORL1 could induce leukemogenesis.

**Figure 2 cancers-17-02443-f002:**
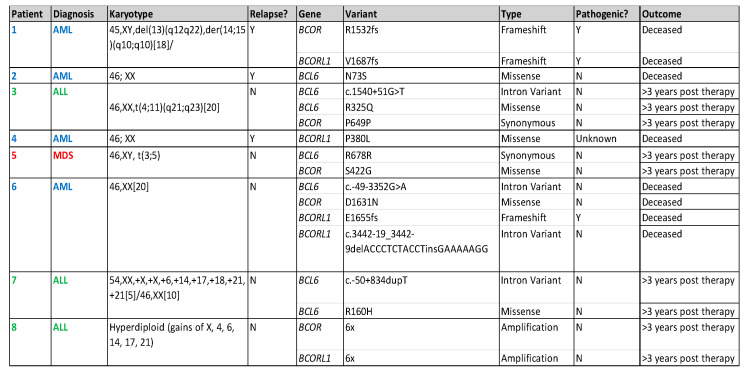
Table describing location of variants and likelihood of pathogenicity for each identified mutation and patient outcomes.

**Figure 3 cancers-17-02443-f003:**
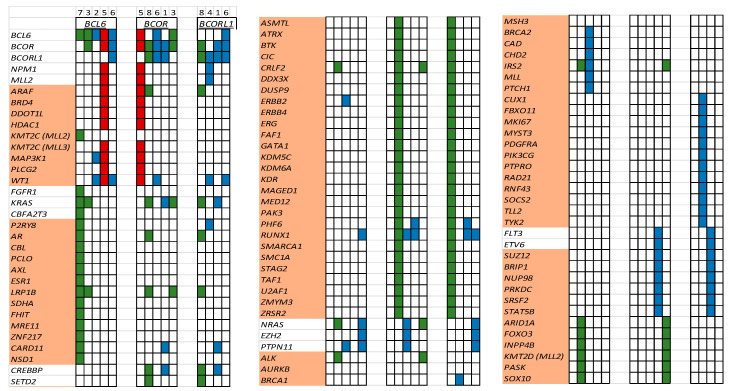
List of co-occurring mutations for each patient (#1–8, red—MDS, green—ALL, blue—AML). Each column depicts one patient. A cell is colored if the corresponding gene (listed in rows) is mutated. Genes highlighted in orange are variants of unknown significance (VUSs).

**Figure 4 cancers-17-02443-f004:**
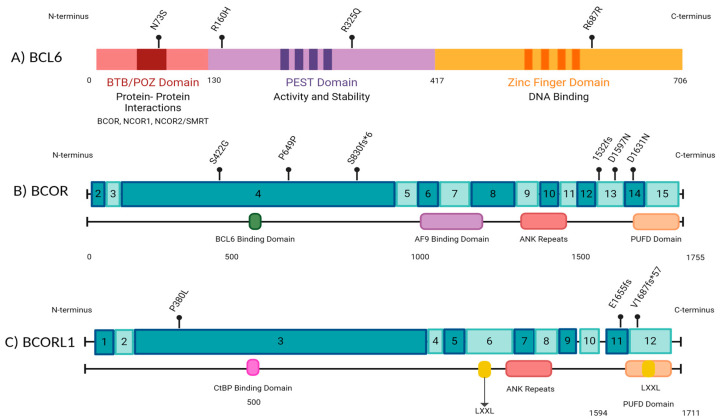
A schematic representation of BCL6 and BCOR gene structures along with the locations of mutations identified by next-generation sequencing (NGS) in our cohort of pediatric leukemia patients. The numbers in the boxes indicate exons. Made with BioRender.com.

## Data Availability

The original contributions presented in this study are included in the article/Appendix A. Further inquiries can be directed to the corresponding authors.

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
