# Peer review of "BCOR, BCORL1, and BCL6 Mutations in Pediatric Leukemias"

_cancers, 2025, doi:10.3390/cancers17152443_

Round 1
Reviewer 1 Report
Comments and Suggestions for Authors
Provides useful insight into the frequeny and nature of BCOR/BCORL1/BCL6 mutations in pediatric leukemias.
The authors evaluated NSG data from 102 pediatric ALL and 82 with MDS/AML and find that the incidence of BCL6/BCOR/BCORL1 mutations is low (4.3%) consistent with findings with adults. They provide information with regards to these mutations, whether they are expected to be pathogenic, and additional mutations seen in other genes; interestingly RUNX1 mutations are common, c/w co-existence of BCOR/BCORL1, BCL6 mutations in patients with familial RUNX1 mutation and AML.
My main concern is the utility of the Kaplan-Meier plot in Fig 2, both due to the low patient number (n=5 for AML) and that some of these patients have non-pathogenic BCOR mutations. I suggest instead providing the outcomes (e.g., MRD after induction, relapse, long-term survival) individually for the 8 patients listed in Table 1A.
Author Response
My main concern is the utility of the Kaplan-Meier plot in Fig 2, both due to the low patient number (n=5 for AML) and that some of these patients have non-pathogenic BCOR mutations. I suggest instead providing the outcomes (e.g., MRD after induction, relapse, long-term survival) individually for the 8 patients listed in Table 1A.
- Modified Table 1A and altered to “Table 1” to include survival outcomes described as alive >2 years post therapy or deceased
- Figure 2 changed to Figure 4 and moved to supplemental material
Reviewer 2 Report
Comments and Suggestions for Authors
BCOR and BCORL1 Mutations in Pediatric Leukemias
The authors retrospectively reviewed patients with pediatric AML and MDS who had BCOR or BCORL1 variants. The manuscript is well written in detail, but there are several points that need improvement.
Major comments:
1) Table 1B: This is a Figure, not a Table. Please correct it to Figure 2.
2)Does the current Figure 2 show RFS or OS? If it is RFS, the Y-axis needs to be corrected to RFS, and if it is OS, the OS should also be mentioned in the main text.
3) For comparison purposes, it is recommended to also present OS and PFS for the entire AML/MDS cohort and OS and PFS for the cohort excluding AML/MDS with BCOR/BCORL1 variants.
Minor comments:
1) Cancers’s “Instructions for Authors” state the following:
References: In the text, reference numbers should be placed in square brackets [ ], and placed before the punctuation; for example [1], [1–3] or [1,3].
The reference numbers appear to be placed after the punctuation marks. Please correct all instances where this occurs.
2) Once again, a comment regarding the citation numbers.
For example,
Page 2 line 46: “genes.[4, [8,9]” should be corrected to “genes [4,8,9].”
Page 2 line 48-49: “leukemogenesis. [10[[11,12].” should be corrected to “leukemogenesis [10-12].”
Carefully review the entire manuscript and make any necessary corrections.
Author Response
Comments and Suggestions for Authors
The authors retrospectively reviewed patients with pediatric AML and MDS who had BCOR or BCORL1 variants. The manuscript is well written in detail, but there are several points that need improvement.
Major comments:
- Table 1B: This is a Figure, not a Please correct it to Figure 2.
- Modified and changed Table 1B to Figure 2
2)Does the current Figure 2 show RFS or OS? If it is RFS, the Y-axis needs to be corrected to RFS, and if it is OS, the OS should also be mentioned in the main text.
- Modified to relapse free survival (moved to Supplemental Figure)
3) For comparison purposes, it is recommended to also present OS and PFS for the entire AML/MDS cohort and OS and PFS for the cohort excluding AML/MDS with BCOR/BCORL1 variants.
- This data was not collected and is unavailable at this time. However, to address this we have included recently published historical outcomes both in introduction (page 3/line 106 -107 /para 2) and discussion (page 3/line 108-124/para 3-4) for comparison.
Minor comments:
- Cancers’ “Instructions for Authors” state the following:
References: In the text, reference numbers should be placed in square brackets [ ], and placed before the punctuation; for example [1], [1–3] or [1,3].
The reference numbers appear to be placed after the punctuation marks. Please correct all instances where this occurs.
- We updated formatting to address this concern
- Once again, a comment regarding the citation
For example,
Page 2 line 46: “genes.[4, [8,9]” should be corrected to “genes [4,8,9].”
Page 2 line 48-49: “leukemogenesis. [10[[11,12].” should be corrected to “leukemogenesis [10-12].”
Carefully review the entire manuscript and make any necessary corrections.
- We carefully reviewed the manuscript and updated the formatting to address this concern
Reviewer 3 Report
Comments and Suggestions for Authors
This study sheds light on mutations that are genuinely rare in pediatrics. At the same time, much of the current draft still feels like an internal working file and, in places, the text pushes harder on prognostic claims than the tiny cohort can comfortably support.
Major limitations
-
Tiny, mixed cohort (n = 8). Survival inferences drawn from five AML/MDS cases (all transplanted) and three ALL cases lack statistical power and blur disease-specific biology.
-
Uncontrolled confounders. High-risk co-mutations (WT1, RUNX1, complex cytogenetics) and transplant at relapse overshadow any independent effect of BCOR/BCORL1.
-
Survival analysis over-interpreted. A Kaplan–Meier curve based on ≤ 5 events suggests precision that simply is not there.
-
The title promises BCOR/BCORL1, yet BCL6 is lumped in without a clear mechanistic rationale, diluting focus.
Minor limitations
-
Inclusion criteria for variant selection (VAF threshold, VUS filtering) are not stated.
-
Relapse-free survival is defined as “three years off therapy,” but the KM axis runs to five years—needs alignment.
-
The discussion on emerging therapeutics (BCL6 inhibitors, PROTACs, Menin blockers) feels speculative given the sample size.
-
Gene schematic (Figure 3) lacks a scale bar and does not explicitly mark the variants reported here.
Recommendations for the authors
-
Recast the manuscript as a Brief Report / Case Series centred on the five AML/MDS patients with BCOR/BCORL1 variants; remove the KM curve and describe outcomes narratively.
-
Clearly acknowledge that poor survival cannot be attributed to BCOR/BCORL1 per se due to co-mutations and transplant timing.
-
Drop the BCL6 cases—or, if retained, justify their inclusion mechanistically and retitle accordingly.
-
Replace the image-based tables with a single, editable Word table listing diagnosis, cytogenetics, variant details, key co-mutations, treatment, and outcome.
Author Response
Comments and Suggestions for Authors
This study sheds light on mutations that are genuinely rare in pediatrics. At the same time, much of the current draft still feels like an internal working file and, in places, the text pushes harder on prognostic claims than the tiny cohort can comfortably support.
Major limitations – see modifications below
- Tiny, mixed cohort (n = 8). Survival inferences drawn from five AML/MDS cases (all transplanted) and three ALL cases lack statistical power and blur disease- specific biology.
- Uncontrolled confounders. High-risk co-mutations (WT1, RUNX1, complex cytogenetics) and transplant at relapse overshadow any independent effect of BCOR/BCORL1.
- Survival analysis over-interpreted. A Kaplan–Meier curve based on ≤ 5 events suggests precision that simply is not there.
- The title promises BCOR/BCORL1, yet BCL6 is lumped in without a clear mechanistic rationale, diluting focus.
Minor limitations – see below
- Inclusion criteria for variant selection (VAF threshold, VUS filtering) are not
- Relapse-free survival is defined as “three years off therapy,” but the KM axis runs to five years—needs alignment.
- KM curve X-axis is measured in months and now goes to 36 months
- The discussion on emerging therapeutics (BCL6 inhibitors, PROTACs, Menin blockers) feels speculative given the sample size.
Gene schematic (Figure 3) lacks a scale bar and does not explicitly mark the variants reported here.
Recommendations for Authors
Recast the manuscript as Brief Report / Case Series centered on the five AML/MDS patients with BCOR/BCORL1 variants; remove the KM curve and describe outcomes narratively.
- KM curve removed and modified, added as a Supplemental figure
- outcomes added to Table 1 for descriptive purposes
Clearly acknowledge that poor survival cannot be attributed to BCOR/BCORL1 per se due to co-mutations and transplant timing.
- Page 6/Lines 250-251 address this “Due to the limited number of patients in our study, and the rarity of BCOR mutations in pediatric myeloid leukemias, we cannot draw definitive conclusions”
- Also added to discussion that high -risk co-mutations and transplant at relapse are uncontrolled confounders and may overshadow any independent effect of BCOR/BCORL1 and impact on survival, in our study.
Drop the BCL6 cases—or, if retained, justify their inclusion mechanistically and retitle accordingly.
- Retitled without BCL6, and added to discussion that role of bcl-6 needs further investigation (page 6-7/lines 270-299)
Replace the image-based tables with a single, editable Word table listing diagnosis, cytogenetics, variant details, key co-mutations, treatment, and outcome.
- Edited tables (Table 1A à Table 1) and changed Table 1B to Figure to address reviewer 1 and 3
Round 2
Reviewer 1 Report
Comments and Suggestions for Authors
The authors added outcomes for each patient to Table 1, as requested, and moved the KM curve to supplemental data, which is acceptable.
Reviewer 2 Report
Comments and Suggestions for Authors
The authors responded appropriately to the peer review comments.
Reviewer 3 Report
Comments and Suggestions for Authors
The manuscript falls short both scientifically and editorially, and therefore does not yet justify publication.